# Peptide-Drug Conjugates: A New Hope for Cancer Management

**DOI:** 10.3390/molecules27217232

**Published:** 2022-10-25

**Authors:** Vivek P. Chavda, Hetvi K. Solanki, Majid Davidson, Vasso Apostolopoulos, Joanna Bojarska

**Affiliations:** 1Department of Pharmaceutics and Pharmaceutical Technology, L M College of Pharmacy, Ahmedabad 380008, Gujarat, India; hetvi5085@gmail.com; 2Institute for Health and Sport, Victoria University, Melbourne, VIC 3030, Australia; majid.hassanzadeganroudsari@vu.edu.au; 3Immunology Program, Australian Institute for Musculoskeletal Science, Melbourne, VIC 3021, Australia; 4Institute of General and Ecological Chemistry, Faculty of Chemistry, Lodz University of Technology, 116 Zeromskiego Street, 90-924 Lodz, Poland

**Keywords:** short peptides, peptide-drug conjugates, drug delivery, cancer

## Abstract

Cancer remains the leading cause of death worldwide despite advances in treatment options for patients. As such, safe and effective therapeutics are required. Short peptides provide advantages to be used in cancer management due to their unique properties, amazing versatility, and progress in biotechnology to overcome peptide limitations. Several appealing peptide-based therapeutic strategies have been developed. Here, we provide an overview of peptide conjugates, the better equivalents of antibody-drug conjugates, as the next generation of drugs for required precise targeting, enhanced cellular permeability, improved drug selectivity, and reduced toxicity for the efficient treatment of cancers. We discuss the basic components of drug conjugates and their release action, including the release of cytotoxins from the linker. We also present peptide-drug conjugates under different stages of clinical development as well as regulatory and other challenges.

## 1. Introduction

Cancer remains one of the major diseases with a high mortality rate despite advances in enhanced diagnostics and treatments. In women, the leading cancer is breast cancer, and in men, it is prostate cancer [1]. At the genetic level, cellular DNA is altered, leading to abnormal gene expression patterns [2]. Consequently, the effects of normal genes which control normal cellular functions such as growth and survival of the cell and invasion/motility, are accentuated [3]. Accumulation of mutations is the main mechanism of alteration. However, non-mutational(epigenetic) changes occur through the DNA methylation process, which is recognized as central to the process [4]. Primarily, cancer can be treated by various approaches including surgery, chemotherapy, radiotherapy, and immunotherapy [5]. Chemotherapeutic drugs generally have a low capacity to penetrate the parenchyma of solid tumors, hence, improvements are vital [6,7]. In addition, the overexpression of cancer associated antigens allows targeted approaches for drugs to be delivered as antibody–drug conjugates or peptide-drug conjugates [8,9], with minimal effects on healthy cells [10].

As we know, communication between cells and their cell surface proteins is required for their survival. In a single organism, numerous cells are connected to form an interactome. The interactome is an extensive network system composed of protein–protein interactions(PPIs) and an array of molecular interactions [11]. Hydrophobic effects hold hydrogen bonding and electrostatic forces together. PPIs networks are formed via the correlation between the human interactome and 130,000–600,000 PPIs. These PPIs are necessary for chemical biology and medicinal chemistry, and modulate signaling pathways and the cells’ systemic functions. Some cellular processes such as transcription, translation, transduction, and replication are catalyzed by PPIs [12]. In living cells, PPIs and proteins play a role in functional building blocks. Diseases such as cancer, infectious and neurodegenerative diseases, and disturbances in cell homeostasis can be caused by errors in a central node/hub of the PPIs network [13]. For the development of new drugs and novel diagnostics, PPIs are attractive targets and have efficient therapeutic potential [14,15].

Peptides, as constituents of proteins, are promising, safe, and effective anti-cancer therapeutics and are engaged in nearly all biological functions. Short peptides do not induce an undesired autoimmune response; they have high selectivity and specificity, are able to cross cell membranes, and penetrate tumors more efficiently [16,17,18,19,20,21,22,23,24,25,26,27]. Herein, applications of peptide-drug conjugates, their basic components, conjugation chemistry, release action, and different stages of their clinical development, regulatory procedures, and other challenges are presented.

## 2. Peptide-Drug Conjugates

In peptide-drug approaches for cancer, it is important to identify a target for cancer cells and a method to target the drug to such cells [28]. Peptides facilitate the penetration of drugs into the body. A chemical modification of peptide and protein drugs improves their enzymatic stability and/or membrane penetration of peptides and proteins [29]. Thus, much focus has been on tumor-targeting peptides. For example, cancer cells, and the tripeptide arginyl-glycyl-aspartic acid (RGD) express adhesion receptors, αvβ3, and αvβ5 integrins-peptides selectively bind to these integrins, allowing drug delivery [30]. In the parenchyma of the tumor, peptides show limited loading capacity but can deliver drugs, biologics, viruses, and nanoparticles to blood vessels successfully [31]. While the molecular targets in non-targeted approaches are not clear, natural and synthetic molecules are screened based on their ability to produce a sufficient phenotypic change in cells, including depletion of mitotic and motility defects and initiation of apoptosis. As such, the final hit compounds are selected based on efficacy to restrict the desired phenotypic properties with minimal side effects [32].

Currently, peptides are used for several diseases, representing a powerful class of medicines [16,22]. Short peptides are advantageous over biologics and small molecules, as they can interact with unknown targets, have a straightforward design, are relatively cheaper to synthesize, and have enhanced tissue penetration [33]. Peptides show excellent performance in transporting molecules to desired targets. Thus, such advantages lead to the identification of several naturally occurring and modified peptides, in addition to their combination with other systems, such as peptide–nanoparticles and peptide-drug conjugates [34]. Countless strategies are available for identifying peptides with anti-cancer activity for targeted and non-targeted approaches. The non-targeted approaches are evaluated from natural sources such as plant extracts, algae, toxins, animal venom, microbes, or screening of synthetic libraries. A targeted approach uses a direct evolution technique for rationally designed peptides (combined principles of biology with materials science) for selected targets [35,36]. Peptide-drug conjugates (PDCs) are composed of different components, which include a homing peptide or device, cytotoxic payload, and a linker that works synergistically to deliver cytotoxins to the targeted receptor on cancer cells (Figure 1) [37]. Each component’s role and mechanisms of action are important considerations when assembling PDCs [38].

An example of PDCs includes the use of gemcitabine, a chemotherapeutic drug. Gemcitabine is attached to a glutaryl linker by an ester bond, which is then attached to the peptide ([D-Lys^6^]-GnRH) (Gonadotropin hormone-releasing hormone (GnRH)) by an amide bond [37]. Notably, PDCs have much broader applications and advantages over antibody–drug conjugates. They are smaller, and consequently, have a better profile in terms of cellular uptake, stability, selectivity, immunogenicity, cellular permeability, tissue penetration, and lower manufacturing costs [39]. However, unlike antibody–drug conjugates, PDCs are not fast-paced in the market due to the poor intrinsic pharmacokinetic properties of the peptides [40]. PDCs have a shorter half-life of minutes, whilst antibody–drug conjugates have a half-life of days to weeks [41]. However, new methodologies are being developed to overcome such factors. Thus far, two PDCs are available in the market, (i) Lu^177^-dotatate (somatostatin as homing peptide conjugates with a radio-therapeutic agent) used for the treatment of gastroenteropancreatic neuroendocrine tumors [42], and (ii) melphalan flufenamide (melflufen), an allogeneic first-in-class peptide-drug conjugate, approved by the US Food and Drug Administration (USFDA) in 2021 for the treatment of multiple myeloma [42], with several others in various phases of development (Table 1).

Another example of a PDC involves the 12-mer exacyclic mimetic peptide(H-FCDGFYACYKDV-OH) of the heavy chain 3 of trastuzumab (a recombinant anti-human epidermal growth factor 2 (HER2) receptor antibody) known as AHNP, which shows high affinity and similar potency as trastuzumab regarding binding affinity (REF). In addition, AHNP linked to the chemotherapeutic drug, doxorubicin, forms an ester bond with the matrix metalloproteinase (MMP-2) cleavage (which is linked to AHNP via an amide bond) [68]. In addition, via carbamate linkage of lysine amino groups between the peptide and drugs, several PDCs targeting somatostatin receptors (SSTR) and other receptors have been designed [69]. However, PDCs with ester linkers have been shown to release the drug faster compared to PDCs with a carbamate linkage [43].

## 3. Components of Peptide-Drug Conjugates

Several approaches have been explored for the selective delivery of effector molecules to cancer cells. PDCs are composed of three main components, including homing peptide, linker, and payload (Figure 1).

### 3.1. Homing Peptide

The first component of PDCs is the carrier or the homing peptide, which eases tumor targeting [40]. Several biologics, including antibodies, proteins, peptides, and small molecules have been investigated in addition to aptamers to facilitate tumor selectivity [70]. The homing peptide is a selected peptide that has a specificity for targeting molecules to specific protein receptors expressed on cancer cells, and often displays high affinity to the target receptor site [71]. Thus, the homing peptide directs the PDC at the desired targeted cell or tissue (as they have enhanced binding affinity for targets) and limits off-target delivery. The binding affinity of homing peptides is an important feature, and it depends on the secondary structure of homing devices such as random coil, β-sheet, and α-helix. This allows greater binding affinity through enhanced stabilization of the secondary structure of the homing peptide [72]. However, this approach has certain disadvantages, such as chemical instability, degradation by enzymes, and fast renal clearance. However, it is possible to gain control over such types of drawbacks by developing modifications of the peptide such as cyclization techniques or non-natural amino acids. Cyclization is achieved either via side chain to side chain, head-tail chain to side chain, or head-to-tail modifications, which are more stable and active compared to their linear counterparts, as seen in various studies [73,74,75,76,77,78]. To produce peptide-drug conjugates, several homing peptides have been used (Table 2).

Peptides can be used as therapeutic agents and as drug carriers [79], but the criteria for action and selection of peptides are quite different. Herein, we discuss all the aspects of peptides. They are used in biotechnology for diagnostic and therapeutic purposes [80]. As an example, calcitonin gene-related peptide is used for migraines, bombesin is used in prostate cancer, somatropin pegol is used in growth hormone deficiency, and BPI-3016 is used in type-2 diabetes as a therapeutic agent [81]. In recent years, peptides have demonstrated improved binding affinity and specificity for challenging binding interfaces due to their smaller size and balance of conformational rigidity and flexibility. There needs to be an understanding and characterization of peptide–protein recognition mechanisms to develop new peptide-based strategies to interfere with endogenous protein interactions or to improve the affinity and specificity of existing approaches. It has been shown that peptide therapeutics can be rationally developed using computational-aided rational design methods. As such, over 60 peptides in clinical trials have been approved worldwide [82]. Delivering biologically active molecules into cells with peptide carriers is one of the most promising approaches. In addition, cancer and infectious diseases could benefit from protein/peptide delivery for efficient and safer vaccines and therapies, as has been shown in models of cancer and multiple sclerosis (REF) [94,95,96,97,98,99]. In recent years, PTD-mediated transfection has demonstrated that future treatments can benefit from ‘protein therapy’ [100]. The blood-brain barrier limits the transport of most therapeutic compounds to the brain in several conditions, such as brain cancers and other neurological disorders. It was previously shown that a synthetic peptide carrier, K16ApoE, resembled the ligand-receptor system that enabled protein transport into the brain. Cisplatin, methotrexate, and other drugs can be delivered non-covalently to the brain with the peptide carrier, by generating transient blood brain barrier permeability [83]. In addition, the interaction between S protein and hACE2 can be potently inhibited by several peptide-based inhibitors, which can be useful for the treatment of COVID-19(REF) [101,102].

In the same way, lactoferrin, an antiviral peptide, is an important drug candidate against COVID-19. Phase II human clinical trials are in progress for RhACE2-APN01, a drug candidate based on recombinant hACE2 peptide. Peptide-based therapeutics for the treatment of COVID-19 include SBP1; Spikeplug target SARS-CoV-2 RBD, EK1; EK1C4; IBP01 and 2019-nCoV-HR1P & HR2P target protein-HR1 [84].

### 3.2. Linker

In peptide-drug conjugations, linkers play a crucial role by linking small drug molecules to peptides and maintaining the structural integrity of PDCs [103]. Based on the mechanisms of PDC, the linker should be selected based on its stability within the circulation to prevent non-specific and premature releasing of the drug at receptor sites, which may result in adverse effects [104]. As such, several conditions are considered when determining which linker to use including pH, enzyme responsiveness, redox responsiveness, and non-covalent linkers (Figure 2) [105].

#### 3.2.1. pH-Responsive Linkers

pH-responsive linkers are the most widely used, and low pH of the tumor microenvironment, intracellular vesicles and organelles, and inflammatory sites play a central role in the release of drugs from PDCs [106]. pH-sensitive linkers which form imine, cis-aconityl, hydrazone, and ketal/acetal bonds are used for drug conjugations [107]. Due to their straightforward reaction between carbonyl groups (aldehydes or ketones) and amines or hydrazides under mild acid conditions, amine and hydrazone linkers are most used. Materials that are capable of releasing their payloads in acidic conditions, such as tumors or infection sites, have been designed using various chemical functionalities including acetal, ortho ester, amine, and hydrazone. Hydrazone linkages are important synthons for a variety of transformations, and they have acquired prominence in pharmaceutical sciences due to their many biological and clinical uses [108]. A series of six molecules comprising 18 different molecules are designed, synthesized, and studied to investigate the effect of structural differences on the degradation kinetics of acetal- and ketal-based linkers [109]. Thus, PDCs have some limitations for targeted drug delivery as the difference [110] in pH between normal and tumor cells may not be sufficient [106].

#### 3.2.2. Enzyme Responsive Linkers

In the progressive early stages of diseases such as cancer, proteolytic enzymes including proteases can affect their role [111]. This involves the progression of the disease by cleavage of biologically important molecules into the cellular microenvironment [112]. Further, their stability is enhanced by adding several alkyl groups such as methyl, adjacent to disulfide bonds. As an example, cathepsin-B or lysosomal cysteine proteases papain regulate the adaptive and innate immune systems, which are unregulated in the tumor microenvironment of some types of cancers [113]. Non-invasive protease activity imaging in vivo with protease-specific cleavable agents offers unique possibilities for the monitoring of disease states. A polymeric carrier with a fluorescent reporter is used to develop protease-sensitive activatable ligands [114,115,116].

#### 3.2.3. Redox Responsive Linkers

In some PDCs, redox sensitivity is provided by the presence of disulfide-containing linkers. The stability of disulfide linkers can be enhanced by adding electron-donating groups such as methyl groups, while the introduction of aromatic groups can lead to bond dissociation [117]. When glutathione levels increase to 1–10 mM/L, it reduces in the cytosol of tumor cells, whereas in the plasma, glutathione is stable at 5 µM/L [118]. This strategy relies on the higher concentration of reducing molecules such as glutathione in the cytoplasm (1–10 mM/L) [119]. In tumor cells, glutathione levels are four times higher than that of normal cells, enabling the selection of an antitumorigenic PDC linker [120]. Diselenide bonds have recently developed as a novel reduction-sensitive linkage. However, its reduced sensitivity has not been observed systemically [121].

#### 3.2.4. Non-Covalent Interaction Linkers

A strong ionic bond between a carboxylic group and metal is widely employed in nanomedicines. Some platinum-based PDCs are in clinical trials in which the metal complexation method is used as the alkylating agent [122]. Metal complexation by using PDCs is used as a potent anti-cancer therapy as a result of their unique properties such as redox activity, variable coordination modes, and reactivity toward organic substrates; metal complexes have attracted a great deal of attention as cancer therapy agents, and have shown to be effective in apoptosis-inducing phase lock loop-based conjugates (removes differences in frequency and phase among input and output signals) [123]. The use of PDCs as metal complexation catalysts is a potent anti-cancer therapy for PLL-based conjugates that induce apoptosis. The simultaneous action of this example suppressed cell migration and invasion on intracellular Ca^2+^ homeostasis perturbation and synergetic apoptosis [124]. As a result, metal complexation with PDCs, RadProtect^®^ (in phase I clinical evaluation, also known as CC-AMI, developed by Original BioMedicals, Taipei, Taiwan) has also been used to treat acute radiation syndrome. PEG-b-PGA micelles connected by ferrous ions chelate the -COOH and -SPO_3_H of amifostine, releasing the drug into the bloodstream with controlled release [125,126]. In addition, the linker’s fate factor (denotes a substance’s total persistence) is considered to demonstrate the drug metabolism effects on the body, and hence resulting toxicity is formed [127]. An ideal linker for selective killing would release the drug only after the intracellular uptake of the conjugates via cancer cells [37]. Upon decomposition of PDCs, cancer cells absorb whatever remains (intact PDCs) for intracellular delivery of the drug. In the synthesis process of drug conjugates, linkers form two chemical bonds: one bond is formed between the peptide and the linker, and another bond is formed between the drug and the linker [128]. A non-cleavable linker is triggered by environmental changes of diseased tissue stimuli such as tripeptide glutathione, or other enzymes present in the tumor microenvironment, and change in pH [103,129].

For improvement in the efficacy of PDCs, the chemistry of the linker plays an important role. Non-cleavable linkers are less preferred over cleavable ones, even though they are more stable [130]. Nevertheless, the linkers should specifically and efficiently release the drug once the targets activate the pharmacological effect (Table 3). Thus, it is obvious to study their chemical effects in the plasma for desirable choice of linkers according to peptide and drug molecules. For example, hydrazones are mainly unstable under acidic conditions (pH 4.5–6.0), and in conditions greater than pH 6.0 the linker is not completely stable in the plasma [131]. Esters are commonly preferred for the conjugation of PDCs due to their ability to be cleaved by the esterase enzyme under acidic conditions. Even though esters do not provide much stability in the plasma, they are still used as a medicinal agent [131]. Carbamates also function similarly to that of esters with relatively higher stability in the plasma. Amides are preferred for the extended duration of action along with the increased therapeutic index due to the release of an active form of conjugated small molecules in the biological system, as amide bonds help and facilitate them in the lysosome with the help of multiple protease enzymes [103,131,132]. In addition, dipeptide linkers are considered, such as Val-Cit and related derivatives if there is no cleavage, as they are cleaved by intracellular proteases [133]. Similarly, tripeptides such as Glu-Val-Cit have been utilized to further enhance the efficacy and stability over dipeptide linkers. Initially, the stability and efficacy of dipeptides and tripeptides were evaluated in mice. However, it was noted that peptide linkers had higher stability in human plasma compared to mouse plasma. This is due to the cleavage of extracellular carboxyl-esterase enzymes in mice, and translation errors may occur when comparing preclinical and clinical data. Thus, the amide bond linker confers suitable plasma stability [133,134]. β-glucuronide is another example of a cleavable linker and has a high advantage of easy drug release, aqueous solubility, and high serum stability [135]. Disulfide linkers are also involved in the formation of peptide conjugates with higher plasma stability. Thus, to achieve the appropriate targeted delivery, hydrazones, esters, carbamates, amides, dipeptides, tripeptides, β-glucuronide, and disulfide bonds provide a diverse set of linkers that are preferably utilized in the development of PDCs (Table 3) [136].

### 3.3. Payloads

The main and third component of PDCs is the drug itself, which can persuade a variety of biological functions. However, in the treatment of cancer, radionuclides and cytotoxic molecules are widely used [137].

#### 3.3.1. Radionuclides

Radionuclides are the second most important type of payload in PDCs. It can be used for two main purposes, that of diagnosis or in therapeutic settings. Various types of PDCs such as [^18^F] Galacto-RGD, ^68^Ga-labeled bombesin analog (RM2) and ^99^mTc-labeled [Phe^7^, Pro^34^] NPY have been investigated under clinical settings as molecular imaging agents. Most commonly, indium-111(^111^In), lutetium-177(^177^Lu), and yttrium-90(^90^Y) are used as radionuclides [138].

#### 3.3.2. Chemotherapeutic Agents

Based on the general mechanism of action, cytotoxic agents are classified as the first group of cytotoxic agents which interfere with the DNA protein complexes/cellular DNA and lead to apoptosis. Anthracyclines such as daunorubicin (DAU) or doxorubicin (DOX) and camptothecin (CPT) are widely used in PDCs. The second group of agents act by inhibiting the biosynthesis of DNA. It is a class of antimetabolites. Folate derivative methotrexate (MTX), nucleoside analog of deoxycytidine, and gemcitabine are widely used. The third group of agents act on microtubules with anti-mitotic abilities. Vinka alkaloid analogs and paclitaxel (PTX) belong to this class of agents [138].

There are thousands of anti-tumor agents available and a wide range of cytotoxic drugs on the market, and each of them come with different limitations including poor pharmacokinetic properties and generation of toxicity to nearby healthy cells whilst targeting an affected one [139]. To overcome these unwanted effects, cytotoxins are attached to peptides with the reduced dose, since peptides provide a specific targeted delivery, and hence the greater proportion of the drug being reached to the specific target site. There are certain criteria for the selection of suitable payloads with respect to PDCs including potency, release through a linker, and stability within the circulation. The payloads are usually chosen with lower IC_50_ within the nM range due to some cytotoxic agents being more toxic, and as such, picometer range is used. The common payloads which are used include daunorubicin, doxorubicin, taxol, and Gemzar [40,140].

## 4. Delivery of Peptide–Drug-Conjugates to the Desired Targeted Tissue

When delivering drugs to cancer cells, the primary goal is to minimize their impact on healthy tissues and ensure that a sufficient amount of the drug reaches the target. This is achieved using two main approaches, both of which aim to alter the pharmacokinetic properties of the drug [141]; (i) use of a delivery vehicle such as nanoparticles, which encapsulate the drug and can dictate its biodistribution through its physicochemical properties, and (ii) to covalently modify the drug with a small moiety that masks the drug’s bioactivity or limits its pharmacokinetic properties [43]. Using peptides, PDCs can have a great degree of functionality, as they can be controlled physiochemically and actively target a particular receptor on cancer cells [142]. The peptides in PDCs are generally short in length and biodegradable, to avoid unwanted immune reactions [143]. By combining different amino acids, a wide variety of PDCs are easily prepared. The sequence controls both the hydrophobicity and the ionization of the conjugate, which affects bioavailability in vitro and in vivo [144].

The physiological action of PDCs depends on several factors including the homing peptide and the linker. Cancer-targeted therapeutic approaches rely on cell surface receptors because they provide specific targeting properties for tumors. To achieve sufficient selectivity, cell surface receptor overexpression is required. It is usually desirable to have a normal cell-to-tumor expression ratio above 1:3 [145]. As discussed, the linker prevents premature release or unwanted release of cytotoxins that can be responsible for the adverse effects. Thus, it is important to gain control over the release of cytotoxins from the linker [146]. A cleavable linker should be cleaved in the presence of a suitable pH or enzymes. Two alternates are available; in the first approach, PDCs follow a similar kind of action to that of antibody-drug conjugates such as the first internalization and then release of a cytotoxin from the linker by cleavage, while the second approach includes cleavage release of a cytotoxin outside the cell followed by internalization [147]. Homing peptides play an important role in targeting the PDCs at specific receptors as they can be a non-penetrating or penetrating peptide. Generally, non-penetrating homing peptides bind with the targeted receptor site and initiate receptor-mediated internalization and endocytosis [148]. PDCs are transported from early endosomes to late endosomes and enter lysosomes, where the pH is low and which contain specific enzymes, where the PDC is cleaved and releases the cytotoxin [132]. As a result, peptide-binding receptors are potential drug carriers with tumor-specific properties due to the properties associated with their activating peptide ligands. Additionally, small molecules or antibodies can also be used to target peptide-binding receptors. As such, the drug compound can be covalently linked to the receptor-binding molecule, making it possible to deliver the drug in a targeted manner. Furthermore, the main demerits are fast renal clearance and poor stability of PDCs in circulation [149]. To overcome these drawbacks, gold nanoparticles and conjugation to antibody Fc or albumin have been used together with the PDCs due to their longer circulation half-life, reliable physicochemical properties, and safety properties which enhance the overall stability of PDCs [150]. One of the greatest advantages of PDCs is that they may overcome/bypass the development of drug resistance in cancer cells. As such, branched cell penetration of peptide-drug conjugates have been studied to overcome drug resistance. Many valuable anti-cancer drugs have undesirable side effects in the current clinical setting, including drug resistance and inefficient cellular uptake. There are several strategies for overcoming limitations in therapeutic research, such as the construction of high-affinity multivalent ligands for drug delivery or selective tumor targeting of chemotherapeutics [151]. Cell-penetrating peptides are particularly effective delivery vehicles, and conjugates of these peptides with anti-cancer medicines have been used to improve cellular uptake. They concluded that in comparison to their linear analogue and doxorubicin, new branched doxorubicin–oligoarginine conjugates demonstrated superior efficacy against wild-type and resistant neuroblastoma tumor cell lines. For the first time, these novel conjugates confirm the effect of high local concentrations on cellular uptake and cytotoxicity in resistant cells [152]. Radiotherapy is mostly used as a treatment of cancer, but it increases normal tissue damage, so it is conjugated with peptides. Peptides linked to radiosensitizing monomethyl auristatin E selectively stimulates CD8^+^ T cells and are reliant on long-term tumor control and immunological memory in irradiated cells. Monomethyl auristatin-E sculpts the tumor immune infiltration in conjunction with ionizing radiation to enhance immune checkpoint inhibition. This is how drug resistance is overcome by PDCs, which will be beneficial for treating trimodal cancers [153]. Efficiency of PDCs depends on the targeted receptor, the pathway of receptor-mediated-endocytosis, and intracellular trafficking. For the treatment of cancer, several cancer-specific receptors or markers are targeted by PDCs such as the integrin (αvβ3) receptor for ovarian cancer, EGFR receptor and MUC1 (CD227) for lung, breast, bladder, and ovarian cancer, NPY(Hy1R) receptor for breast cancer and Ewing sarcoma, and MC1R receptor for melanoma (REFS) [154,155,156]. By the targeted drug delivery approach, side-effects can be reduced, and the efficacy of the drug can be increased [38]. Trafficking of extracellular macromolecules into cells (endocytosis) and across cells (transcytosis) is facilitated by receptor-mediated transport mechanisms. During this process, ligands bind to specific cell-surface receptors, cluster together within endocytic vesicles, and sort into their respective vesicles [157]. Internalization and trafficking of CSPG-bound recombinant VAR2CSA lectins are examples and used in cancer treatment [158].

## 5. Applications

PDCs have a broad range of applications (Table 4). They can be applied against viruses such as SARS-CoV-2 [159,160] or multi-resistant bacteria [161]. PDCs also have relevance in the development of efficient vaccines. CoVac501, a self-adjuvanting peptide vaccine conjugated to a toll-like receptor (TLR)-7 agonist is a promising therapeutic against SARS-CoV-2 [162] with antifungal applications [163]. Likewise, in cancer, a self-adjuvanting multicomponent vaccine combining per-glycosylated MUC1 peptide and TLR2 agonist Pam3CysSer shows strong antibody generation in animal models (REF) [164]. Moving forward, peptide conjugates with a fluorescent dye can be used in diagnostic applications i.e., cancer imaging. In addition, polyanionic peptides conjugated with anti-inflammatory drugs are delivered into bones and other tumor cells [165]. Peptide conjugates acting on the central nervous system [166] or in gene delivery [167] are promising.

### 5.1. Anti-Cancer Therapy

In the context of applications of conjugated peptides for anti-cancer therapy, apoptosis and tumor accumulation-related peptides as well as the role of peptide-based nanotechnology, are discussed.

### 5.2. Apoptosis

Peptide-inducing apoptosis in tumor cells include Ra-V (deoxybouvardin) against breast cancer [180], and Hodgkin lymphoma [181]. DEVD (Asp-Glu-Val-Asp)-monomethyl auristatin-E conjugate shows strong therapeutic effects with minimal toxicity. More specifically, self-assembled nanoparticles deliver large amounts of monomethyl auristatin-E [182]. Self-assembled nanoparticles are advantageous as they show diversity in their chemical makeup, are biocompatible, enable high loading capacity of peptides and/or drugs, and can target sites on cancer cells. In fact, the peptide KLA (acetyl-KLAKLAK)_2_-NH_2_) conjugated with a tumor-homing peptide iRGD (CRGDKGPDC) has improved penetration of low-grade tumor tissue and cells with high selectivity and low systemic toxicity [42,183].

### 5.3. Tumor Accumulation

Diverse techniques for targeting tumors with peptides have been introduced. For example, non-RGD (arginine-glycine-aspartate) disulfide(SS)-bridged cyclo-peptide (ALOS-4) has specificity for integrin avβ3, overexpressed on several cancers including human metastatic melanoma. ALOS-4 conjugated with a topoisomerase I inhibitor, camptothecin, revealed increased cytotoxicity in human metastatic melanoma cells and decreased cytotoxicity in normal cells [184]. The tetra-branched peptide NT4 conjugated with paclitaxel binds to tumor membrane-sulfated glycosaminoglycans with strong selectivity showing more effective tumor regression [185]. Dimeric 123B9 peptide, targeting EphA2 (related to cancer metastasis, overexpressed in melanomas, ovarian, prostate, lung, and breast cancers), conjugated with paclitaxel inhibits lung metastasis in breast cancer models [186]. A protein-G-derived albumin-binding domain (ABD) conjugated with doxorubicin via a pH-sensitive linker demonstrates a longer half-life in the plasma and four times higher accumulation within the tumor [182]. Albumin-binding peptide (DICLPRWGCLW)-based bioconjugates are stable complexes for tumor-targeting effect [42,182].

### 5.4. Cancer Immunotherapy

The treatment of cancer can be possible by controlling the function of immune cells [42]. Programmed cell death ligand (PD-L1) is overexpressed on several cancer cells [187,188,189,190]. Interaction with programmed cell death protein 1 (PD-1; CD279), overexpressed on activated T cells, enables immune evasion of cancer cells. Inhibiting PD-1/PD-L1 interactions is a promising therapeutic approach. D-PPA [NYSKPTDRQYHF], a PD-L1-binding peptide, is an immune-related peptide (immunomodulating peptide). Interestingly, the hydrophilic D-type polypeptide (D-PPA) does not show cytotoxicity, but it can inhibit tumor growth and prolong survival by blocking the PD-1/PD-L1 interaction. In comparison to the equivalent L-peptide, D-peptides improve enzymatic stability, increase plasma half-life, improve oral bioavailability, binding activity, and specificity with the receptor of target proteins [191]. PD-L1-binding nanoparticles can be conjugated with doxorubicin leading to minimal toxicity and enhanced internalization [192]. Melittin peptide [GIGAVLKVLTTGLPALISWIKRKRQQ] can be conjugated with IL-2, improving the antitumor effect, and reducing side effects [193]. Macrophages play an essential role in innate and adaptive immunity, such that M2pep [Cys-YEQDPWGVKWWY] can be modified to increase specificity to target M2 tumor-associated macrophages. Co-encapsulation of CSF-1R siRNA and M2pep can target M2 tumor-associated macrophages for reprogramming to the M1 type, resulting in increased anti-tumor immune responses. Consequently, synergistically therapeutic effects in pancreatic cancer cells have been noted [194].

### 5.5. Nanotechnology

Peptides provide different advantages as drugs such as low toxicity, potency, and specificity. However, several practical challenges exist, such as short half-life, poor stability, and susceptibility to protease digestion. Nanotechnology has shown great potential in overcoming many traditional challenges of medicines. By using nano-drug delivery systems, numerous traditional pharmaceuticals have successfully passed into clinical trials and clinics. The combination of nanotechnology and peptides in nanomedicine offers excellent opportunities for developing novel, safe, and more efficient treatments against many diseases, including different types of cancers. Nanotechnology is an appealing strategy for maintaining functional forms of peptide biomolecules (biorecognition, signal transduction for treatment, toxicity) that otherwise could be degraded by digestive enzymes. Conjugation of peptides to nanoparticles enhances the stability and cellular uptake, leading to excellent efficacy and reduced toxicity. Protease-activatable liposomes were one of the earliest combinations of nanotechnology and peptides. It was designed by conjugating an acetylated dipeptide substrate (N-Ac-AA) through its carboxyl terminus to the amino group of 1,2-dioleoyl-sn-glycero-3-phosphoethanolamine (DOPE) to promote cellular internalization [195]. The resulting (N-Ac-AA-DOPE), together with dioleoyl trimethylammonium propane (DOTAP) and phosphatidyl-ethanolamine (PE), was assembled into non-fusogenic liposomes. The nanoparticles were activated by elastase or proteinase K to become fusogenic to improve intracellular delivery. Since then, tumor-targeting peptides have been successfully incorporated in nano-carriers that deliver small-molecule drugs, targeted imaging agents, liposomes, oligonucleotides, and inorganic nanoparticles to tumors. Moreover, peptide-related nanoparticles, such as peptidomimetic self-assembled nanoparticles and peptide aptamers, have also shown promising results in targeted drug delivery for cancer treatment [196,197].

The cell-penetrating peptide, KT_2_(NGVQPKYKWWKWWKKWW-NH_2_), conjugated to gold nanoparticles against breast cancer is a good example [198]. Notably, peptide-based bioconjugates can form nanoparticles by self-assembly resulting in increased permeability and retention effects at the tumor site [199,200]. Curcumin, a polyphenol derived from Curcuma longa(turmeric) root, has relevance either in terms of targeting or nano-formulation. The encapsulation of curcumin with albumin overcomes shortcomings of curcumin (low water solubility and bioavailability). In addition, the conjugation of nanoparticles with RK-10 (GSGSGSTYLCGAISLAPKAQIKESL), a PDL1-targeting peptide, leads to increased cytotoxicity against breast cancer [201]. Self-assembling peptides can be used as scaffolds for cell regeneration or carriers for the delivery of drugs and can be utilized to target the controlled release of stable drugs with less toxicity [202]. Moreover, studies showed tumor lymphatic vessel-targeting peptides present a novel approach for targeted drug delivery for cancer treatment [203].

Recently, the number of targeting peptide–nanoconjugate drugs has significantly increased [204,205]. Using nanotechnology, new approaches are available to combine targeting peptides with other functional peptides and various diagnostic or therapeutic molecules. Peptide targeting itself is becoming combinatorial with the use of more than one peptide in a single entity, the use of peptides that also serve as proteolytic substrates to provide more function [206]. The further development of self-assembly peptide-based nanotechnology can promise highly effective anti-cancer therapeutic options [42]. Figure 3 presents different types of peptides which nanoparticles can deliver and provide a more efficient therapeutic impact for cancer treatment.

### 5.6. Bicycle–Toxin Conjugates

A bicycle peptide is a new modality consisting of 9–20 amino acids, including three cysteine residues reacting with a linker to form a rigid conformation of the peptide [207]. A bicycle can act as a transporter of drugs by bicycle-toxin conjugates. This approach offers some advantages such as slower renal clearance, deeper tumor penetration, and rapid extravasation [207].

### 5.7. Peptide–Dendrimer Conjugates

Peptide dendrimers, covalent (type I and II), built from (un)natural amino acids, and noncovalent (type III) that are not attached to multiple frameworks through the periphery, have a highly branched construction and unique properties-thermal performance, viscosity, and use for encapsulation. In the drug delivery mechanism, the drug is attached by (non)covalent (encapsulation)/bond. Peptide dendrimers have good biocompatibility and tunable amino acid features [208]. They can selectively deliver doxorubicin to the target cancer site [209]. Polyethylene glycol (PEG) incorporated into the dendrimer leads to an increased half-life. In the case of breast cancer, a dendrimer nanoparticle with doxorubicin is efficiently reduces tumor size [209]. Peptide dendrimers can also be used in the treatment of colorectal cancer with controlled gemcitabine drug delivery [210,211]. Further, dendrimers have also been shown to deliver cancer peptides and proteins to the immune system for effective stimulation and activation for immunotherapeutic approaches [212,213].

### 5.8. Cell-Penetrating Peptides

Membrane translocating peptides, such as penetratin from Antennapedia with sequence RQIKIWFQNRRMKWKK when tandemly linked to cancer peptides can stimulate anti-cancer immune responses [214,215,216,217]. The mechanisms by which cell-penetrating peptides enter cells and delivery of drugs, peptides, proteins, and DNA, is via a combination of energy-independent (membrane fusion) and energy-dependent pathways (endocytosis). Some peptides are also fully energy-independent in their uptake [218]. Cell-penetrating peptides are arginine-rich peptides, and due to their positive charge, they interact with negatively charged drug molecules and cell membranes through non-covalent interaction, including electrostatic interactions. As such, the application of these arginine-rich peptides has been used to deliver biosensors, drugs, and vaccines and allows entry through the blood-brain barrier [219]. An example is the use of these peptides linked to chemotherapeutic drugs, such as doxorubicin, for its specific delivery to triple-negative breast cancers [220].

## 6. Human Clinical Trials

We note that over twenty PDCs are currently in human clinical trials (some are summarized in Table 5 at different stages of clinical development). Zoptarelin against metastatic endometrial cancer [221,222], polyglutamic acid-paclitaxel conjugate for therapy of non-small cell lung cancer [223], and Vitafolid, a folic acid-vinca alkaloid conjugated with a peptide moiety against epithelial ovarian cancer [224], which reached phase III trial, are good examples [225].

In addition, appealing next-generation bicycle–toxin and peptide–dendrimer conjugates are in the pipeline, and their diverse variations are undergoing clinical trials [211]. BT1718, BT5528, and BT8009 [226] are worth mentioning. BT1718 consists of the specific targeting bicycle conjugated to the MD1 cytotoxin via a cleavable disulfide bond. Its target is 1 matrix metalloproteinase related to bladder, lung, breast, ovarian, and endometrial cancer. Phase I of clinical trials revealed good tolerability and stability and allowed the progression to Phase II [226]. BT5528 is formed from bicycle conjugation, via an enzyme cleavable Val-Cit linker, to an MMAE payload. It targets Ephrin type A receptor 2 related to several tumors and tyrosine kinase, which is responsible for cell adhesion, migration, proliferation, and differentiation. Phase I/II showed well tolerability [207,226]. BT8009, like BT5528 (but with a Val-Cit cleavable linker), targets Nectin-4, overexpressed in breast, bladder, lung, or pancreatic cancer. This bicyclic toxin conjugate presented promising results, including limited off-target toxicity in clinical trials [210,211]. Furthermore, it could be highlighted that nanotechnology offers excellent potential for PDCs to obtain clinical trials. The nanostructures of modified drugs, such as camptothecins, 10-hydroxycamptothecin, hydroxychloroquine, paclitaxel, doxorubicin, and dexamethasone via conjugation with peptides can be successfully used to treat a plethora of cancers, overcoming the drawbacks of drugs and enhancing their efficacy and efficiency [43,69,227]. The self-assembled PDCs, despite the last progress, have not been accepted either by the FDA or for clinical trials so far [227]. Nevertheless, in the coming years, rapid approval of anti-cancer PDCs is expected, thanks to the high investment in clinical trials by big Pharmaceutical Corporations (Pfizer, Oncopeptides, Pepgen Corporation, Bicycle Therapeutics, AstraZeneca).

## 7. Challenges for the Delivery of Peptide-Drug Conjugates: The Way Forward

Despite the remarkable progress in the research of PDCs, they still face challenges such as poor stability, rapid renal clearance [228], and translation of their favorable features into clinical outcomes for patients. PDCs often fail in clinical trials due to difficulty in translating rational drug conjugate designs into effective anti-cancer therapies [44]. The question of how and when PDCs are is released after entering the cell remains open. Thus, PDCs must be designed and synthesized more efficiently. Notably, fluorescent probes (such as fluorogenic with aggregation-induced emission feature) can be incorporated into PDCs to monitor this process [43]. Another problem is the oral administration of peptides as a new strategy for utilizing potential PDCs in human clinical trials. In this context, a new self-assembly method to form pectindiydroartemis in hydroxycamptothecin nanoparticles to PDC delivery by the oral route is worth mentioning. It could improve drug loading and release [209,229].

## 8. Conclusions

Peptide-drug conjugates are a great promise for cancer management, with two compounds, Melluflen and Lu^177^-dotatate, on the market and many others, such as peptide–dendrimer or bicycle–toxin conjugates, undergoing clinical trials. PDCs are a next-generation of targeted therapeutic agents as the equivalent antibody-drug conjugates, offering many advantages such as enhanced tumor penetration, selective delivery of cytotoxic payloads to target cells, improved efficacy, multifunctionality, biodegradability, reduced immunogenicity, and systemic toxicity, simplicity, easy structural modification, and lower costs of synthesis. Nevertheless, there are still challenges to overcome such as stability, bioactivity, and renal clearance. On the other hand, new innovative technologies such as nanotechnology are a promising prospect for potentiating anti-cancer efficacy, with potentially fewer side effects in terms of PDCs, thus driving the development of the field.

## Figures and Tables

**Figure 1 molecules-27-07232-f001:**
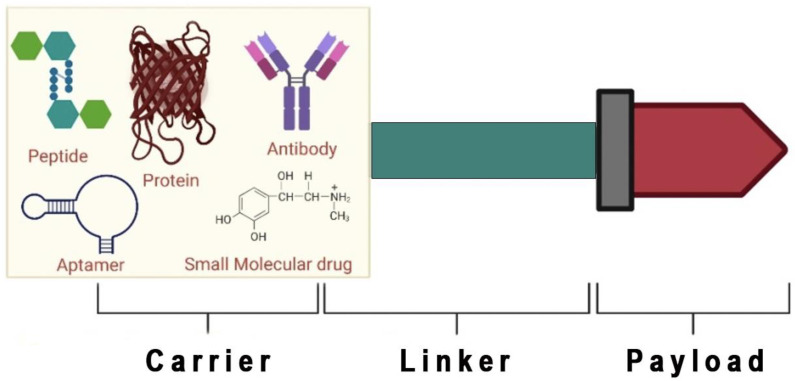
Schematic diagram of the components of peptide-drug conjugates (Created with Biorender.Com). Peptide, protein, antibody, small molecular drug, and aptamer is used as a carrier. If a peptide is used as a carrier, it is known as a peptide-drug conjugate.

**Figure 2 molecules-27-07232-f002:**
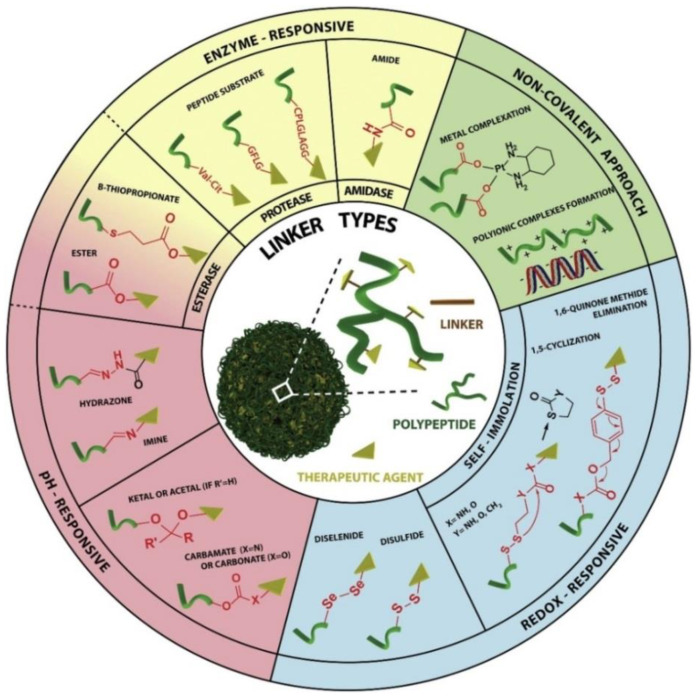
Different types of polypeptide-drug linkers commonly used in the rational design of poly peptide-drug conjugates (Adopted under CC BY 4.0 from [103]). In the figure, white bubble indicates PDCs. The green triangle is a therapeutic agent. The red colored line indicates the linker and the green colored ribbon type structure indicates polypeptide. In the linker, different types of linkers are discussed.

**Figure 3 molecules-27-07232-f003:**
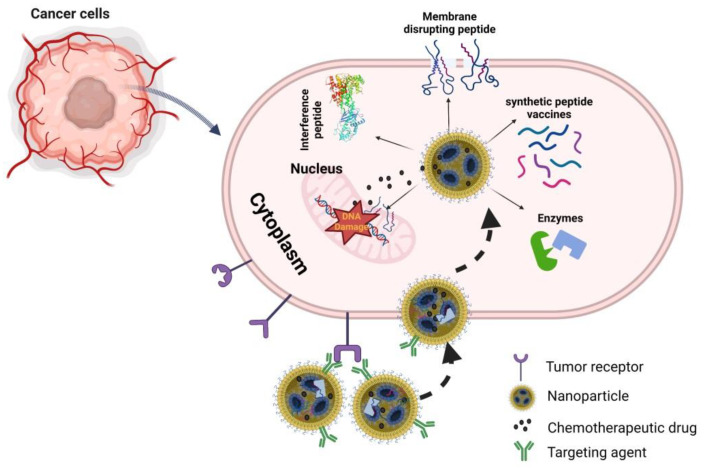
Summary of different types of peptides that nanoparticles can deliver to cancer cells. Figure was created using biorender.com.

**Table 1 molecules-27-07232-t001:** Some peptide-drug conjugates against cancer in different developmental phases.

Development Phase	Peptide-Drug-Conjugates			References
US FDA approved	Lu^177^ dotatate	Peptide	Somatostatin analogue	[43,44,45,46]
		Drug	Radio-therapeutic agent	
		Linker	Amide (chelating agent Lu^177^ chelates with meta-chelating agent DOTA)	
		Action	Somatostatin receptor-2 mediated delivery of nucleotide	
		Indication	Neuroendocrine cancer	
	Melflufen (melphalan flufenamide) amino peptide	Peptide		[47,48]
		Drug		
		Linker	Enzymatically cleaved linker	
		Action	Targets aminopeptidases. Rapidly releases alkylating agent into cancer cells	
		Indication	Multiple myeloma, ovarian cancer, breast cancer, acute myeloid leukemia, hematologic malignancies, and solid tumors	
Clinical development	Paclitaxel with Angiopep-2	Peptide	Angiopep-2	[49,50]
		Drug	Paclitaxel	
		Linker	Ester	
		Action	LPR-1 mediated brain uptake	
		Indication	Brain cancer	
	Paclitaxel with Poliglumex	Peptide	Poliglumex	[51,52,53]
		Drug	Paclitaxel	
		Linker	Ester	
		Action	Prolongs the exposure of the tumour to the active form of drug by minimizing the systemic enhancing the permeability of tumor vasculature	
		Indication	To treat a wide variety of cancers	
	Thapsigargin with Tetrapeptide	Peptide	Tetrapeptide	[52,53]
		Drug	Thapsigargin	
		Linker	Ester	
		Action	Tumor activation (extracellularly)	
		Indication	To treat a wide variety of cancers	
	Doxorubicin-GnRH	Peptide	GnRH	[54]
		Drug	Doxorubicin	
		Linker	Ester	
		Action	GnRH-mediated delivery to cancer cells	
		Indication	Ovarian and endometrial cancer	
	Maytansinoid with Bicyclic peptide	Peptide	Bicyclic peptide	[55]
		Drug	Maytansinoid	
		Linker	Disulfide	
		Action	Membrane type-1 matrixmetalloproteinase delivery to toxin material	
		Indication	Cancer	
	Doxorubicin-Tetrapeptide	Peptide	Tetrapeptide	[56,57]
		Drug	Doxorubicin	
		Linker	Amide	
		Action	Tumor activation (extracellularly)	
		Indication	Cancer	
	^123^I-Glutamate urea lysine	Peptide	Glutamate urea lysine	[58]
		Drug	Radiotherapeutic agent (^123^I)	
		Linker	Amide	
		Action	Prostate-specific membrane antigen-mediated delivery of nucleotide	
		Indication	Specifically used for prostate cancer	
	^111^In-DTPA-D-Phe-1-octreotide	Peptide	D-phe-1-octreotide	[59]
		Drug	^111^In-DTPA	
		Linker	Amido bond	
		Action	Diagnosis of neuroendocrine tumors (somatostatin receptor-positive tumors) using SPECT or planar scintigraphy	
		Indication	Diagnostic Radiology	
	Aminopeptides N (CD13) peptide-hTumour Necrosis Factor (NGR-TNF)	Peptide	NGR	[60]
		Drug	hTNFα	
		Linker	Amido bond	
		Action	Targets angiogenic tumor blood vessels (vasculature), through the NGR motif	
		Indication	Elapsed ovarian cancer	
	Paclitaxel-Poliglumex conjugate CT2103 (Opaxio™)	Peptide	Poliglumex	[61]
		Drug	Paclitaxel (chemotherapeutic drug)	
		Linker	Ester bond	
		Action	Paclitaxel therapeutic index is restricted due to poor tumor exposure and high systemic exposure. Paclitaxel poliglumex (PPX) combination increases tumor exposure by taking advantage of tumor tissue’s hyperpermeable vasculature and reduced lymphatic clearance. The release of paclitaxel is dependent, at least in part, on PPX metabolism by the lysosomal protease cathepsin B, which is increased in many tumor types.	
		Indication	Metastatic breast cancer	
	Paclitaxel linked to brain delivery vector angiopep-2 (ANG1005)	Peptide	Angiopep-2	[62]
		Drug	Paclitaxel	
		Linker	Ester bond	
		Action	Paclitaxel is blocked from attaining its target in malignant gliomas by the presence of the efflux pump P-glycoprotein (P-gp) at the blood-brain barrier. Angiopep-2 peptide vectors have the potential to improve their efficacy in the treatment of brain tumors.	
		Indication	Recurrent malignant glioma	
	Natural luteinizing hormone-releasing hormone (LHRH) ligand linked to a cationic membrane disrupting peptide (EP-100)	Peptide	LHRH	[63]
		Drug	CLIP71	
		Linker	Amino bond	
		Action	EP-100 interacts with the negatively charged membrane upon accumulation on the cell membrane via LHRH receptor targeting, resulting in lysis and cell death.	
		Indication	LHRH-receptor-expressing solid tumors	
Preclinical development	Gemcitabine conjugation to an integrin binding knotting peptide-Ecbellium elaterium trypsin inhibitor (EETI)-2.5Z, via Val-Ala-PABC linker (EETI-2.5Z-Val-Ala-PABC-genciabine)	Peptide	Knotting peptide	[64,65]
		Drug	Gemcitabine	
		Linker	Amide, ester, cathepsin-B, carbamate	
		Action	DNA replication block assisted by integrin knotting peptide that specifically allows tumor delivery	
		Indication	Breast cancer, ovarian cancer, geo-blastoma, and pancreatic cancer	
	DOTA (bifunctional chelating ligand; 1,4,7,10-tetraazacyclododecane-1,4,7,10-tetraacetic acid) linked to bombesin (BBN) (DOTA-GABA-BBN)	Peptide	Bombesin 7–14, from frog skin	[66]
		Drug	DOTA	
		Linker	g-aminobutyric acid	
		Action	Bombesin is overexpressed on cancer cells. DOTA-GABA-BBN is a specific radioligand for gastrin-releasing peptide-positive tumors. Used in targeted PET imaging	
		Indication	Used for diagnostic radiology	
	MTX-YTA2	Peptide	Acetyl-YTAIAWVKAFIRKLRK -amide	[67]
		Drug	Methotrexate (MTX)	
		Linker	Amino bond	
		Action	Cell-penetrating peptide (YTA2) conjugated to MTX as therapeutic for drug-resistant cancer cells	
		Indication	Breast cancer	

**Table 2 molecules-27-07232-t002:** Homing peptides used for the development of Peptide-drug-conjugates.

Homing Peptide	Receptor Site [71,73,73,79,80,81,82,83,84,85,86,87,88,89,90,91,92,93]
Somatostatin (SST)	Somatostatin receptor
Gonadotropin-releasing hormone (GnRH)	Receptor version of hormone-GnRH-R
Angiopep-2	Low-density lipoprotein receptor-related protein (LRP-1)
Epidermal growth factor (EGF)	Epidermal growth factor receptor (EGFR): HER1, HER2, HER3, HER4
REG	Integrin αvβ3 receptor
iREG	Integrin αvβ3/αvβ5 receptor
GE11	Epidermal growth factor receptor (EGFR): ErbB1
D-Lys^6^-LHRH	Luteinizing hormone-releasing hormone receptor (LHRH-R)
Octreotide	Somatostatin receptor 2/5 (SSTR2/5)

**Table 3 molecules-27-07232-t003:** Different types of linkers used in peptide-drug conjugates.

Type of Linker	Name	Comment
pH-responsive	Acetal/Ketal bond	An acidic environment in endosomes
	Hydrazone bond	
	Vinyl ether bond	
Enzyme responsive	Gly-Phe-Leu-Gly	Cathepsin B
	Phe-Lys	
	Val-Citrulline	
Redox responsive	Disulfide bond	Reduction in endosomes by glutathione
Non-covalent interaction	Amido bond	The potential of modifying the particle’s surface peptide layer to stabilize the particle for systemic in vivo injection or for targeting opens fascinating possibilities for improving cargo trafficking
	Carbon chain	
	Ether bond	

**Table 4 molecules-27-07232-t004:** Selected applications of peptide-drugconjugates.

Peptide	Drug/Vaccine	Indications	Reference
Somatostatin peptide analogue	^177^Lu-dotate	Gastroenteropancreatic neuroendocrine tumors	[168]
Di-amino acid	Melflufen	Multiple myeloma	[169]
Arginine-glycine-aspartic acid	Naproxen/Ibuprofen	Anti-inflammatory/additives therapy with anti-cancer drugs to provide specific targeted delivery	[170]
Asparagine-glycine-arginine	Naproxen/ Ibuprofen		
Arginine-glycine-aspartic acid/Asparagine-glycine-arginine	Ketoprofen	Anti-cancer therapy	[171]
AE147 peptide	Docetaxel	Advanced therapy for metastatic tumor	[172]
Gonadotropin-releasing hormone I/II	Daunorubicin	Targeted tumor therapy with high binding affinity and provides apoptotic effects	[173]
Dimethylmaleic anhydride	Doxorubicin	Targeted tumor therapy	[174]
Lysine	Neomycin-B	Antibiotics mainly evaluate their activity against gram-positive bacteria	[175]
Disulfide	Kanamycin	Successfully shows the activity in *Mycobacterium* tuberculosis	[176]
Polylactide-co-glycolide	Pyridopyrimidine derivatives	Anti-tubercular activity	[177]
Dipeptide	Cinnamic acid	Anti-protozoal agent	[178]
Ubiquicidin	Coumarin derivatives	Anti-fungal agent	[179]
Self-adjuvanting peptide vaccine conjugated toll-like receptor 7 agonists	CoVac501	SARS-CoV-2 and other infectious diseases	[162]

**Table 5 molecules-27-07232-t005:** Various PDCs under different stages of clinical development.

Name of PDCs	TTP	Payload	Linker	Indication	Development Phase	Clinical Trials Registry
ANG1005	Angiopep-2	Paclitaxel	Succinic acid	Leptomeningeal disease from breast cancer	Phase III	NCT02048059 NCT03613181
NGR015, NGR-hTNF	CNGRCG (1,5SS)	Human TNF	Amide	Malignant pleural mesothelioma	Phase III	NCT01098266 NCT03804866 NCT00484276 NCT00483080
[^18^F]-AlF-NOTA-octreotide	Octreotide	^18^F	NOTA	PET or GEP-NETs	Phase II/III	NCT04552847
BT1718	MT1-MMP binder	DM1	Disulfide	Solid tumors	Phase II	NCT03486730
[^18^F]-Fluciclatide	RGD	^18^F	PEG	PET imaging	Phase II	NCT00918281 NCT00565721 NCT01176500 NCT01633255 NCT01788280
[^18^F]-RGD-K_5_	Cyclo-(RGDfK)	^18^F	NOTA	PET imaging	Phase II	NCT00988936 NCT02490891 NCT03364270
G-202 (mipsagargin)	DγEγEγEγE	Thapsigargin	Amide	Solid tumors	Phase II	NCT02381236 NCT02067156 NCT02607553 NCT01777594 NCT01056029
68Ga-NODAGA-E[cyclo(RGDyK)] 2	E[cyclo(RGDyK)]2	^68^Ga	NODAGA	PET imaging	Phase II	NCT03445884 NCT03271281
PEN-221	fCYwKTCC (2,7 SS)	DM-1	Disulfide	Neuroendocrine and small cell lung cancers	Phase II	NCT02936323
^90^Y-DOTATOC	3 Tyr-octreotat	^90^Y	DOTA	PRRT	Phase II	NCT03273712
GRN1005	Trevatide	Paclitaxel	Ester	Non-small Cell Lung Cancer (NSCLC) With Brain Metastases	Phase II	NCT01497665
BT5528	EphA2 binder	MMAE	Amide	EphA2-positive NSCLC	Phase I/II	NCT04180371
68Ga-NOTA-BBN-RGD	cyclo(RGDyK) and BBN	68Ga	NOTA	PET/CT and PET imaging	Phase I	NCT02749019 NCT02747290
tTF-NG	GNGRAHA	tTF	Amide	Malignant solid tumors	Phase I	NCT02902237
TH1902	Docetaxel peptide–drug conjugate	Alexa^488^	Cleavable ester linker	Solid Tumor, TNBC-Triple-Negative Breast Cancer, Hormone Receptor-positive Breast Cancer, Epithelial Ovarian Cancer, Endometrial Cancer, Cutaneous Melanoma, Thyroid Cancer, Small-cell Lung Cancer, Prostate Cancer	Phase I	NCT04706962
MB1707	CXC	Paclitaxel	Ester	Advanced Solid Tumor, Breast Cancer, Non-Small Cell Lung Cancer, Ovary Cancer, Pancreatic Neoplasms	Early Phase I	NCT05465590

**Abbreviations:** AlF, aluminum fluoride; AMBF3, ammonimethyltrifluoroborate; DγEγEγEγE, Asp-γ-(Glu)4-OH; DM-1, a maytansine derivative; DOTA, cyclododecyltetraacetic acid; f, D-Phe; 18F, fluorine-18; 68Ga, gallium-68; GEP-NETs, gastroenteropancreatic neuroendocrine tumors; MMAE, monomethyl auristatin E; NODAGA, 1,4,7-triazacyclononane,1-glutaric acid-4,7-acetic acid; NOTA, 1,4,7-triazacyclononane-1,4-7-triacetic acid; NSCLC, non-small cell lung cancer; SCN-Bz, 2-pisothiocyanatobenzyl; SN-38, 7-ethyl-10-hydroxycamptothecin; TATE, [3 Tyr]octreotate; TTP, tumor-targeting peptide; w, D-Trp; y, D-Tyr; ^90^Y, yttrium-90. bNCT numbers: ClinicalTrials.gov, USA.

## Data Availability

Not applicable.

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
