# Peer review of "Peptide-Drug Conjugates: A New Hope for Cancer Management"

_molecules, 2022, doi:10.3390/molecules27217232_

Round 1
Reviewer 1 Report
Overview
This manuscript is yet another review on the current developmental status of peptide drug conjugates (PDCs). the authors plan to review the advantages of using small peptides rather than for instance larger antibodies to deliver cytotoxic drugs to cancer cells, as well as the regulatory challenges facing PDCs as they progress into the clinic.
Comments
- The manuscript is well researched and thorough. However, it should be noted that from Jan 1, 2021 through to Aug 31, 2022, the Web of Science database has cataloged 183 reviews on PDCs! Therefore, to be noticed, this manuscript should contribute something new to the filed. I did not find anything unique in the document.
- The manuscript is somewhat jumbled in content. In particular, the use of peptides as therapeutic agents or as drug carriers are discussed together. In effect, the action and selection criteria for these two types of products are quite different and should be discussed separately.
- There are least 2 important areas not discussed that might overcome the problem in point 1 above. A) It is suggested that one advantage of PDCs is that they may overcome/bypass the development of drug resistance in cancer cells. B) Part of the efficiency of PDCs depends on the targeted receptor, the pathway of receptor-mediated-endocytosis and intracellular trafficking.
Author Response
Dear Referee,
thank you very much. I am sending our responses:
Review 1:
„This manuscript is yet another review on the current developmental status of peptide drug conjugates (PDCs). the authors plan to review the advantages of using small peptides rather than for instance larger antibodies to deliver cytotoxic drugs to cancer cells, as well as the regulatory challenges facing PDCs as they progress into the clinic”.
Comments
- The manuscript is well researched and thorough. However, it should be noted that from Jan 1, 2021 through to Aug 31, 2022, the Web of Science database has cataloged 183 reviews on PDCs! Therefore, to be noticed, this manuscript should contribute something new to the filed. I did not find anything unique in the document.
Response: Thank you for your valuable comment. We have discussed comment 3 in section 4 and revised the manuscript to bring the novelty part as expected.
- The manuscript is somewhat jumbled in content. In particular, the use of peptides as therapeutic agents or as drug carriers are discussed together. In effect, the action and selection criteria for these two types of products are quite different and should be discussed separately.
Response: Thank you for the valuable comment. Peptides can be used as therapeutic agents and as drug carriers, but the criteria for the action and selection of peptides are quite different. In this review, we discuss all the aspects of peptides. They are used in biotechnology for diagnostic and therapeutic purposes. We have added clear sentences in most sections to make the distinction clearer
- There are least 2 important areas not discussed that might overcome the problem in point 1 above. A) It is suggested that one advantage of PDCs is that they may overcome/bypass the development of drug resistance in cancer cells. B) Part of the efficiency of PDCs depends on the targeted receptor, the pathway of receptor-mediated-endocytosis and intracellular trafficking.
Response: Thank you for your valuable comment. We have discussed this part in section 4.
Reviewer 2 Report
The authors give an overview over current state of the art of peptide-drug conjugates. It certainly makes sense to compile the many efforts in the field over the last year in a review and the tables the authors generated are rather useful. However, the logic of the manuscript and the writing is so awkward that I am not very confident that the manuscript can be brought into an acceptable form. It is not only the language and logic of many sentences, some statements are moreover wrong or contradictory. The paper has to be totally rewritten, most preferably with the help of a chemist who is a (near) native English speaker. Some obviously problematic statements (these may not be all) are listed below:
Li 67: peptides support the penetration of drugs. The meaning of this sentence is unclear.
Li 79: what is meant with uncharted targets?
Li 88: Whilst in a targeted approach, the direct evolution: Whilst in a targeted approach, the direct evolution technique is used for the generation of peptides for selected targets with good rational design [30-31]. What does “with good rational design” mean in this context?
Figure 1: The figure legend should describe what is depicted . The scheme is not understandable. What are these multicolor components that are associated with the Pacman style module? The figure is totally unclear and has to be optimized.
Li 100: The statement that PDC have an advantage over antibody-drug conjugates with respect to half-life is definitely wrong. ADCs have a half-life of days or weeks, PDC of minutes.
Li113: “A 12-mer peptide (H-FCDGFYACYKDV-OH) obtained from the heavy chain 3 of the trastuzumab (a recombinant anti- human epidermal growth factor 2 (HER2) receptor antibody). This statement is unprecise. The peptide contains two cysteines which are absent in trastuzumab.
Li113: For the peptide ([D-Lys6]-GnRH) the GnRH part is not explained and the 6 most likely should be lowercase and not uppercase.
Li 118 “sensitive linker from matrix metalloproteinase” should read “for matrix-metalloproteinase cleavage”.
Li 111 Why is he example auf a gemcitabine drug conjugate placed here? The whole sections looks misplaced. This is unclear to the reader.
Li 135: What is meant with „other than that of the aptamers“?
Li 134: remove the „;“ in the middle of the sentence
Li 137:Should “provides a strong attraction” mean “displays high affinity”?
Li 139 -142: What the authors likely want to point out is the fact that preformation of secondary structure contributes to enhanced affinity binding to target molecules. The original statement is unclear.
Li 156: Bad Language: “Considerations are considered”
Li 160: “low pH of the tumor microenvironment, intracellular vesicles and organelles, and inflammatory sites play a central role in the uptake of PDCs. “Why should this be the case? Is uptake meant or does this refer to stability? I do not see why pH influences uptake of PDCs. Moreover, what is the chemical structure of such linkers and how is pH dependent drug release programmed in?
Li 169: As an example, cathepsin-B or lysosomal cysteine proteases papain regulate the adaptive and innate immune systems, This is only rather indirectly correct and moreover why should this influence the activity of PDCs? There do exist classical linker peptide sequences such as Val-Cit that are preferably cut by cathepsin B and therefore can be used to release the drug after endosomal uptake but this has nothing to do with immunological issues.
Li 168: THEY affect the role
Li 174: “In the PDCs redox sensitivity is provided by the disulfide linkers” In some PDCs redox sensitivity is provided by the presence of disulfide-containing linkers
Li 175: What does the statement that glutathione “clears in the cytosol of the tumor cells” mean? What is meant with “clears”? Which glutathione, reduced or oxidized? “hence enabling the selection of an anti-tumorigenic DIOSULFIDE-BOND containing PDC linker”
Li 181: “Some Pt-based PDCs are in clinical trials in which the metal complexation method is widely used” Statement not understandably. What has this to fo with linker design? “Metal complexation by using PDCs is used as a potent anti-cancer therapy (HOW AND WHY?)and has shown to be effective in apoptosis-inducing phase lock loop-based (phase lock loop has to be explained!) conjugate”
184 “In addition, the fate factor of a linker is considered to demonstrate the drug metabolism effects on the body” What does the term fate factor mean and how can it demonstrate effect? The whole sentence is non understandable.
Li 186: “would release the drug only after the cancer cell” after??????
Li 187: “However, most of the linkers are cleaved the moment they are in the system (upon systemic administration of the PDC)” Wrong statement, if this wopuld be the case PDCs would not make any sense.
Li 189: “Upon removal of PDC” What is meant with removal? Decomposition?
Li 191: “In the synthesis process of the drug conjugates, linkers form two chemical bonds, one bond is formed between the peptide and the linker, and another bond is formed between the drug and the linker.” Yes is the definition of a linker but what does that sentence want to express? Why does a new paragraph start?
Li 194-205: The hole section is of more general nature. Why is it placed in section named “Non-covalent interaction linkers”
Li 204: What does pronounced mean?
Li 207: Figure 1 should read Figure 2
As to the figure: What is the bubble in the white inlet circle? What are thre green strands with triangles on ?? Moreover there are bonds depicted that are not described in the text such as polyionic complex formation and also diselenide linkers and more.
Li 211: “Non-cleavable linkers are less preferred over cleavable as they have lower stability in the circulation for targeted therapeutics” They have higher stability since they are not cleavable!
Li 213: once the targeted tissue activates the pharmacological effect. Language. The tissue does not activate an effect.
Li 215: “For example, hydrazones are mainly cleavable under acidic conditions (pH4.5-6.0), and conditions greater than pH 6.0 cause the linker to be relatively unstable in the plasma [92].” Language. What is likely meant is that hydrazones ar unstable at acidic pH but also at neutral pH they are not completely stable.
Li 218: Wording. Esters do not cleave peptides. The are cleaved
Li 223: due to the release of an active form of conjugated small molecules in the biological system? What is meant?
Li 224: Language: amide bonds process them. A bond does not process anything
Li 226: if cleavage is not observed. Statement unclear
Li 235. Explain MABC
Li 236: Either the statements on stability of disulfide linkers is moved to the subchapter 3.2.2. redox stable.. or a subchapter non linker stability could be considered.
Li 248: Language: “second utmost group” 251 clinical settling
Li 271: Some cytotoxic agents are amore toxic than nM Ic50 (picomolar range, e.g. MMAE)
Li 295: Language. “a cleavable linker should be cleaved” Yes that is why it is called a cleavable linker.
Li 311: Half-life extension: Not only nanoparticles but also conjugation to antibody Fc or albumin!
Li333: Adcetris cannot be considered as an peptide drug conjugate. It is an ADC!
Li 359: “Conjugation with programmed „ most likely interaction is meant.
Li 360: Language. An inhibition cannot be a target
Li 361: “D-PPA [NYSKPTDRQYHF], a PD-L1-binding peptide, is an immune-related peptide” What means immune-related?
Li 363: What does D-type mean?
Li 444: It is not necessarily a combination of energy-dependent and independent uptake mechanisms. Some peptides are fully energy-independent in their uptake.
Li 463: What is meant with “1 matrix metalloproteinase” Relevant proteases are matriptase 1, MMP2 MMP9 etc.
Li 478: What is more, clinical trials are very expensive. Tricial statement. Remove it.
Li 480: Replace giant by big
Li 493: Language. A potential cannot fail
Li 495: question ‘of how. Remove the ‘
Li 497: What means “in terms” in this context?
Li 498: unclear, which process should be monitored
Li 502: it is unclear why the new self-assembled route described is advantageous and novel.
Li 512: For a balanced review: If the advantages are listed, the dark side should also be listed (stability, renal clearance, PK etc.)
Author Response
Dear Referee,
thank you very much. We are sending our responses:
The authors give an overview over current state of the art of peptide-drug conjugates. It certainly makes sense to compile the many efforts in the field over the last year in a review and the tables the authors generated are rather useful. However, the logic of the manuscript and the writing is so awkward that I am not very confident that the manuscript can be brought into an acceptable form. It is not only the language and logic of many sentences, some statements are moreover wrong or contradictory. The paper has to be totally rewritten, most preferably with the help of a chemist who is a (near) native English speaker. Some obviously problematic statements (these may not be all) are listed below:
Li 67: peptides support the penetration of drugs. The meaning of this sentence is unclear.
Response: Thank you for your valuable comment. We have revised the sentence as Peptides facilitate the penetration of drugs into the body.
Li 79: what is meant with uncharted targets?
Response: Thank you for your valuable comment. Uncharted targets means unknown targets.
Li 88: Whilst in a targeted approach, the direct evolution: Whilst in a targeted approach, the direct evolution technique is used for the generation of peptides for selected targets with good rational design [30-31]. What does “with good rational design” mean in this context?
Response: Thank you for your valuable comment. We have revised it as, A targeted approach uses the direct evolution technique to design peptides that are rationally designed for selected targets. (Good rational design = Joining biology with materials science requires the ability to design, engineer and control biology/solid-state materials interfaces at the molecular level.)
Figure 1: The figure legend should describe what is depicted . The scheme is not understandable. What are these multicolor components that are associated with the Pacman style module? The figure is totally unclear and has to be optimized.
Response: Thank you for your valuable comment. The figure is created with bioreander.com and the components are selected based on the templates available. We have now modified the figure with simple version of the components.
Li 100: The statement that PDC have an advantage over antibody-drug conjugates with respect to half-life is definitely wrong. ADCs have a half-life of days or weeks, PDC of minutes.
Response: Thank you for your valuable comment. After cross verification we have changed the sentence and also provided reference for the same.
Li113: “A 12-mer peptide (H-FCDGFYACYKDV-OH) obtained from the heavy chain 3 of the trastuzumab (a recombinant anti-human epidermal growth factor 2 (HER2) receptor antibody). This statement is unprecise. The peptide contains two cysteines which are absent in trastuzumab.
Response: Thank you for your valuable comment. This is a 12-mer exocyclic mimetic peptide and the sequence is correct, as a mimetic. Reference: https://www.ncbi.nlm.nih.gov/pmc/articles/PMC5812454/. We have included the reference in the article.
Li113: For the peptide ([D-Lys6]-GnRH) the GnRH part is not explained and the 6 most likely should be lowercase and not uppercase.
Response: Thank you for your valuable comment. Gonadotropin hormone-releasing hormone (GnRH) and change the uppercase of 6.
Li 118 “sensitive linker from matrix metalloproteinase” should read “for matrix-metalloproteinase cleavage”.
Response: Thank you for your valuable comment. We have changed the sentence.
In addition, AHNP-linked to the chemotherapeutic drug doxorubicin is formed via an ester bond to the for matrix metalloproteinase (MMP-2) cleavage (which is linked to AHNP via an amide bond)
Li 111 Why is the example of a gemcitabine drug conjugate placed here? The whole sections looks misplaced. This is unclear to the reader.
Response: Thank you for your valuable comment. We have removed that part.
Li 135: What is meant with „other than that of the aptamers“?
Response: Thank you for your valuable comment. We have corrected the sentence. A number of biologics, including antibodies, proteins, peptides, and small molecules, have been investigated in addition to aptamers to facilitate tumor selectivity.
Li 134: remove the „;“ in the middle of the sentence
Response: Thank you for your valuable comment. We have removed it.
Li 137:Should “provides a strong attraction” mean “displays high affinity”?
Response: Thank you for your valuable comment. Yes, it means it displays high affinity.
Li 139 -142: What the authors likely want to point out is the fact that preformation of secondary structure contributes to enhanced affinity binding to target molecules. The original statement is unclear.
Response: Thank you for your valuable comment. We have modify the sentence for better understanding and clarity.
Li 156: Bad Language: “Considerations are considered”
Response: Thank you for your valuable comment. As such, several conditions are considered when determining which linker to use, including pH, enzyme responsiveness, redox responsiveness, and non-covalent linkers. Now may be it is clear.
Li 160: “low pH of the tumor microenvironment, intracellular vesicles and organelles, and inflammatory sites play a central role in the uptake of PDCs. “Why should this be the case? Is uptake meant or does this refer to stability? I do not see why pH influences uptake of PDCs. Moreover, what is the chemical structure of such linkers and how is pH dependent drug release programmed in?
Response: Thank you for your valuable comment. Here pH influences the release of the drug.
Li 169: As an example, cathepsin-B or lysosomal cysteine proteases papain regulate the adaptive and innate immune systems, This is only rather indirectly correct and moreover why should this influence the activity of PDCs? There do exist classical linker peptide sequences such as Val-Cit that are preferably cut by cathepsin B and therefore can be used to release the drug after endosomal uptake but this has nothing to do with immunological issues.
Response: Thank you for your valuable comment.
Li 168: THEY affect the role
Response: Thank you for your valuable comment. We have corrected the sentence. In the progressive early stages of a disease like cancer, proteolytic enzymes such as proteases they affect the role.
Li 174: “In the PDCs redox sensitivity is provided by the disulfide linkers” In some PDCs redox sensitivity is provided by the presence of disulfide-containing linkers
Response: Thank you for your valuable comment. In some PDCs redox sensitivity is provided by the presence of disulfide-containing linkers.
Li 175: What does the statement that glutathione “clears in the cytosol of the tumor cells” mean? What is meant with “clears”? Which glutathione, reduced or oxidized? “hence enabling the selection of an anti-tumorigenic DIOSULFIDE-BOND containing PDC linker”
Response: Thank you for your valuable comment. Now we have corrected the sentence.
Li 181: “Some Pt-based PDCs are in clinical trials in which the metal complexation method is widely used” Statement not understandably. What has this to fo with linker design? “Metal complexation by using PDCs is used as a potent anti-cancer therapy (HOW AND WHY?)and has shown to be effective in apoptosis-inducing phase lock loop-based (phase lock loop has to be explained!) conjugate”
Response: Thank you for your valuable comment. We have corrected the sentence for better meaning. Metal complexation by using PDCs is used as a potent anti-cancer therapy as a result of their unique properties, such as redox activity, variable coordination modes, and reactivity toward organic substrates, metal complexes have attracted a great deal of attention as cancer therapy agents, and has shown to be effective in apoptosis-inducing phase lock loop-based conjugate (removes differences in frequency and phase among input and output signals).
184 “In addition, the fate factor of a linker is considered to demonstrate the drug metabolism effects on the body” What does the term fate factor mean and how can it demonstrate effect? The whole sentence is non understandable.
Response: Thank you for your valuable comment. We have clarified the fate factor in the bracket in the revised manuscript.
Li 186: “would release the drug only after the cancer cell” after??????
Response: Thank you for your valuable comment. We have revised the sentence as below in revised manuscript.
An ideal linker for selective killing would release the drug only after the intracellular uptake of the conjugates via cancer cell.
Li 187: “However, most of the linkers are cleaved the moment they are in the system (upon systemic administration of the PDC)” Wrong statement, if this would be the case PDCs would not make any sense.
Response: Thank you for your valuable comment. We have removed that sentence.
Li 189: “Upon removal of PDC” What is meant with removal? Decomposition?
Response: Thank you for your valuable comment. We have revised the sentence for better meaning.
Li 191: “In the synthesis process of the drug conjugates, linkers form two chemical bonds, one bond is formed between the peptide and the linker, and another bond is formed between the drug and the linker.” Yes is the definition of a linker but what does that sentence want to express? Why does a new paragraph start?
Response: Thank you for your valuable comment. We have made the appropriate modifications.
Li 194-205: The hole section is of more general nature. Why is it placed in section named “Non-covalent interaction linkers”
Response: Thank you for your valuable comment. We have made the changes for the better understanding.
Li 204: What does pronounced mean?
Response: Thank you for your valuable comment. We have modify it for better meaning.
Li 207: Figure 1 should read Figure 2
As to the figure: What is the bubble in the white inlet circle? What are thre green strands with triangles on ?? Moreover there are bonds depicted that are not described in the text such as polyionic complex formation and also diselenide linkers and more.
Response: Thank you for your valuable comment. We have modified the numbering. This is an adopted figure from advanced drug delivery reviews. White bubble indicates PDCs. Green triangle is therapeutic agent. We have discussed all approaches related to the linker in the text.
Li 211: “Non-cleavable linkers are less preferred over cleavable as they have lower stability in the circulation for targeted therapeutics” They have higher stability since they are not cleavable!
Response: Thank you for your valuable comment. Non-cleavable linkers are less preferred over cleavable although it is more stable than the cleavable.
Li 213: once the targeted tissue activates the pharmacological effect. Language. The tissue does not activate an effect.
Response: Thank you for your valuable comment. We have modified the setntence as below.
Nevertheless, the linkers should specifically and efficiently release the drug once the targets activates the pharmacological effect. We have corrected the sentence.
Li 215: “For example, hydrazones are mainly cleavable under acidic conditions (pH4.5-6.0), and conditions greater than pH 6.0 cause the linker to be relatively unstable in the plasma [92].” Language. What is likely meant is that hydrazones ar unstable at acidic pH but also at neutral pH they are not completely stable.
Response: Thank you for your valuable comment. We have modified the sentence as below.
For example, hydrazones are mainly unstable under acidic conditions (pH4.5-6.0), and conditions greater than pH 6.0 the linker is not completely stable in the plasma.
Li 218: Wording. Esters do not cleave peptides. The are cleaved
Response: Thank you for your valuable comment. We have modified the sentence as below.
Esters are commonly preferred for the conjugation of PDCs due to it’s ability to be cleaved by the esterase enzyme under acidic conditions.
Li 223: due to the release of an active form of conjugated small molecules in the biological system? What is meant?
Response: Thank you for your valuable comment. We have corrected the sentence for better understanding.
Li 224: Language: amide bonds process them. A bond does not process anything
Response: Thank you for your valuable comment. We have corrected the sentence for better understanding.
Li 226: if cleavage is not observed. Statement unclear
Response: Thank you for your valuable comment. We have modified the sentence as below.
In addition, dipeptide linkers are considered such as Val-Cit and related derivatives if there is no cleavage.
Li 235. Explain MABC
Response: Thank you for your valuable comment. MABC= Monoclonal Antibody Beta Glucuronidase. This is now clear in the revised paper
Li 236: Either the statements on stability of disulfide linkers is moved to the subchapter 3.2.2. redox stable.. or a subchapter non linker stability could be considered.
Response: Thank you for your valuable comment. We have moved this portion to the subchapter 3.2.2.
Li 248: Language: “second utmost group” 251 clinical settling
Response: Thank you for your valuable comment. We have modified the sentence as below.
Radionuclides are the second most important type of payload in PDCs.
Li 271: Some cytotoxic agents are more toxic than nM Ic50 (picomolar range, e.g. MMAE)
Response: Thank you for your valuable comment. We have modified the sentence as below.
The payloads that are chosen usually with lower IC50 within the nM range because some cytotoxic agents are more toxic than nM IC50 (picometer range, e.g. MMAE).
Li 295: Language. “a cleavable linker should be cleaved” Yes that is why it is called a cleavable linker.
Response: Thank you for your valuable comment. We have modified the sentence as below.
A cleavable linker should be cleaved in the presence of particular pH or enzymes.
Li 311: Half-life extension: Not only nanoparticles but also conjugation to antibody Fc or albumin!
Response: Thank you for your valuable comment.
We have added conjugation to antibody Fc or albumin also.
Li333: Adcetris cannot be considered as a peptide drug conjugate. It is an ADC!
Response: Thank you for your valuable comment. We have removed it.
Li 359: “Conjugation with programmed „ most likely interaction is meant.
Response: Thank you for your valuable comment. Yes, it means interaction.
Li 360: Language. An inhibition cannot be a target
Response: Thank you for your valuable comment. We have corrected the sentence.
Inhibiting PD-1/PD-L1 interactions is a promising therapeutic approach.
Li 361: “D-PPA [NYSKPTDRQYHF], a PD-L1-binding peptide, is an immune-related peptide” What means immune-related?
Response: Thank you for your valuable comment. It means it is immunomodulating peptide.
Li 363: What does D-type mean?
Response: Thank you for your valuable comment. As mentioned on section 5.4 D-type polypeptide is a PD-L1 binding peptide [NYSKPTDRQYHF].
Li 444: It is not necessarily a combination of energy-dependent and independent uptake mechanisms. Some peptides are fully energy-independent in their uptake.
Response: Thank you for your valuable comment. We have modify it as below.
Some peptides are also fully energy-independent in their uptake.
Li 463: What is meant with “1 matrix metalloproteinase” Relevant proteases are matriptase 1, MMP2 MMP9 etc.
Response: Thank you for your valuable comment. Among the six MTMMPs identified to date, MT1, MT2, MT3, and MT5 have a type I transmembrane domain, while MT4 and MT6-MMP are tethered to the plasma membrane by a GPI anchor.(doi https://doi.org/10.1016/S0304-3835(02)00699-7)
Li 478: What is more, clinical trials are very expensive. Tricial statement. Remove it.
Response: Thank you for your valuable comment. We have removed it.
Li 480: Replace giant by big
Response: Thank you for your valuable comment. We have replaced it.
Li 493: Language. A potential cannot fail
Response: Thank you for your valuable comment. We have modified the sentence accordingly.
Li 495: question ‘of how. Remove the ‘
Response: Thank you for your valuable comment. We have removed it.
Li 497: What means “in terms” in this context?
Response: Thank you for your valuable comment. Thus, PDCs must be designed and synthesized in terms more efficiently.
Li 498: unclear, which process should be monitored
Response: Thank you for your valuable comment. Here process refers to how and when PDC is released after entering the cell. This has been made clear in the revised paper
Li 502: it is unclear why the new self-assembled route described is advantageous and novel.
Response: Thank you for your valuable comment. We have modified this section for clarity
Li 512: For a balanced review: If the advantages are listed, the dark side should also be listed (stability, renal clearance, PK etc.)
Response: Thank you for your valuable comment. We have also included the cons as below. On the other hand, there are still challenges to overcome like stability, bioactivity, renal clearance, PK etc.
Round 2
Reviewer 1 Report
The authors have incorporated my suggestions into the revised manuscript.